# Sonocatalytic Degradation of Chrysoidine R Dye Using Ultrasonically Synthesized NiFe₂O₄ Catalyst

**Yogesh M. Gote, Pankaj S. Sinhmar and Parag R. Gogate \***

Department of Chemical Engineering, Institute of Chemical Technology, Matunga, Mumbai 400019, India
* Correspondence: pr.gogate@ictmumbai.edu.in

**Abstract:** The novel ultrasound-assisted co-precipitation method was successfully applied for the synthesis of the NiFe₂O₄ catalyst, which offered the advantages of lower particle size and better crystalline structure without affecting the phase planes. Furthermore, the efficacy of synthesized catalysts was evaluated using ultrasound-assisted catalytic degradation of Chrysoidine R dye. The study was designed to evaluate the effect of different parameters, such as pH, duty cycle, power output, and catalyst loading on Chrysoidine R dye degradation using a 5 wt% NiFe₂O₄ catalyst synthesized ultrasonically. At the optimized condition of 120 W ultrasonic power, 70% duty cycle, 3 pH, 0.5 g/L catalyst loading, and 160 min of reaction time, the best degradation of 45.01% was obtained. At similar conditions, the conventionally synthesized catalyst resulted in about 15% less degradation. Chrysoidine R dye degradation was observed to follow second-order kinetics. To accelerate the degradation, studies were performed using hydrogen peroxide at various loadings where it was elucidated that optimum use of 75 ppm loading showed the maximum degradation of 92.83%, signifying the important role of the co-oxidant in ultrasound-assisted catalytic degradation of Chrysoidine R dye. Overall, the present study clearly demonstrated the potential benefits of ultrasound in catalyst synthesis as well as in catalytic degradation.

**Keywords:** Chrysoidine R dye; ultrasound-assisted synthesis; NiFe₂O₄; sonocatalytic degradation





## 1. Introduction

Wastewater treatment is an essential element of sustainability as natural water resources are limited and purification of saline water is costly. The hazardous effluents discharged by various industrial sectors accelerate the pollution levels and result in contamination of freshwater resources, which has been a hot topic of discussion for over a century. Hazardous-colored wastewater produced by textile enterprises is one of the leading causes of water pollution as most of the color pigments are resistant to bio-degradation, temperature, light, and other chemicals [1]. The removal of these dye compounds is the key issue in effluent treatment for textile-processing companies. Since the traditional wastewater treatment methods often fail to provide sufficient pollutant degradation efficiency, produce secondary pollutants, and are time consuming processes that are not cost effective, new greener and innovative degradation methods have been researched actively lately. One such method is the sonocatalytic degradation approach, which has been reported to actively degrade different dye pollutants, such as methyl orange using MnO₂/CeO₂ [2], anazolene sodium using Dy-doped Cdse [3], methylene blue using CoFe₂O₄/SiO₂/CuMoF [4], polyenes using Se/Fe₂O₄ [5], brilliant green using CaFe₂O₄ [6], and Congo red using Ni-Co-MnO₂ [7] as catalysts. Ultrasound induces the formation, subsequent growth under the compression—rarefaction cycles, and intense collapse of micro-bubbles in an aqueous mixture leading to much higher temperature and pressure pulses, considered as local hot spots. During the cavity collapse, water molecules and oxidants undergo thermal dissociation, mainly generating the hydroxyl radicals and other oxidizing agents, depending on the type

of oxidants used or the components present in the liquid mixture. Ultrasound alone, however, is not an effective approach due to lower degradation efficiency and higher processing times for the target dye molecule, and coupling with a catalyst and oxidant intensifies the degradation based on generation of higher quantum of active radicals, resulting in higher degradation of pollutants present in effluent.

Among the different semiconductor particles that are gathering attention, nickel ferrite is a promising catalyst in the field of catalytic degradation as it possesses a band gap of 1.53 eV [8], offering broad absorption in visible regions, which is beneficial to catalysis applications [9]. $NiFe_2O_4$ can also provide useful catalytic action during the sono-catalytic degradation and for the dissociation of hydrogen peroxide. Considering these aspects, the selection of $NiFe_2O_4$ as the catalyst complex in the current work is justified.

The synthesis of various metal-loaded semiconductors has been reported using various synthesis methods, such as solvo-thermal [10], molten salt synthesis [11], hydrothermal [12], and glycerol-assisted sol-gel [13]. The major demerits of these methods are large crystalline zone, high-temperature requirements, and lack of homogeneity. The simple and easy route of synthesis is chemical co-precipitation [14–16], which offers the benefits of mild operating conditions, but it is often observed that the particle size and catalytic activity are poor though it offers good homogeneity and better crystallinity than other methods. To overcome these demerits, we propose a modified ultrasound-assisted co-precipitation method for nickel ferrite synthesis. In the current study, the sonocatalytic degradation of Chrysoidine R dye has also been investigated using the synthesized nickel ferrite catalyst. The influence of various parameters, such as duty cycle, pH, power, and catalyst loading on the degradation of Chrysoidine R dye were evaluated. The catalytic activity of synthesized catalyst samples was examined under the optimized process conditions. The effect of use of an additional oxidant on sono-catalytic degradation was also investigated. Based on the literature assessment, it can be clearly stated that the degradation of Chrysoidine R dye using an ultrasound-assisted catalytic approach based on ultrasonically synthesized $NiFe_2O_4$ catalyst has not been evaluated, establishing the novelty of the current work.

## 2. Results and Discussions

### 2.1. Characterization of Nickel Ferrite Oxide Catalyst

To examine the potential advantages offered by the $NiFe_2O_4$ catalyst complex obtained using the ultrasound assisted synthesis method compared to that of the conventional method, the samples were characterized using particle size and XRD analysis.

### 2.1.1. Particle Size Analysis

Quantification of the particle size revealed the ultrasonically prepared catalyst complex resulted in a particle size of 15.58 μm, which was found to be approximately 4.31 times smaller than the conventionally synthesized catalyst (80.62 μm), as depicted in Figure 1a,b.

The obtained results clearly indicate that the proposed sonication method can significantly reduce the particle size, hence lowering post-micronizing costs. Use of the ultrasound in the catalyst synthesis generally aids in achieving lower particle size due to the turbulence and shear effects generated during the cavity collapse. Vardikar et al. [17] reported a similar finding of a reduced particle size of 293 nm for the ultrasonically produced KL-CTS-$TiO_2$ catalyst as opposed to a particle size of 439 nm for the conventional approach. Díez-García et al. [18] also reported that $NiTiO_3$ nanorods synthesized under sonication resulted in smaller size (1.8 μm in length and 0.6 μm in dia) nanorods as compared to the conventional approach of synthesis (2.2 μm in length and 0.7 μm in dia). Dalvi et al. [19] also observed a significant reduction in particle size of Fe-doped $TiO_2$ catalyst from 806.4 μm to 31.22 μm on shifting a conventional approach to the ultrasound-assisted synthesis approach.

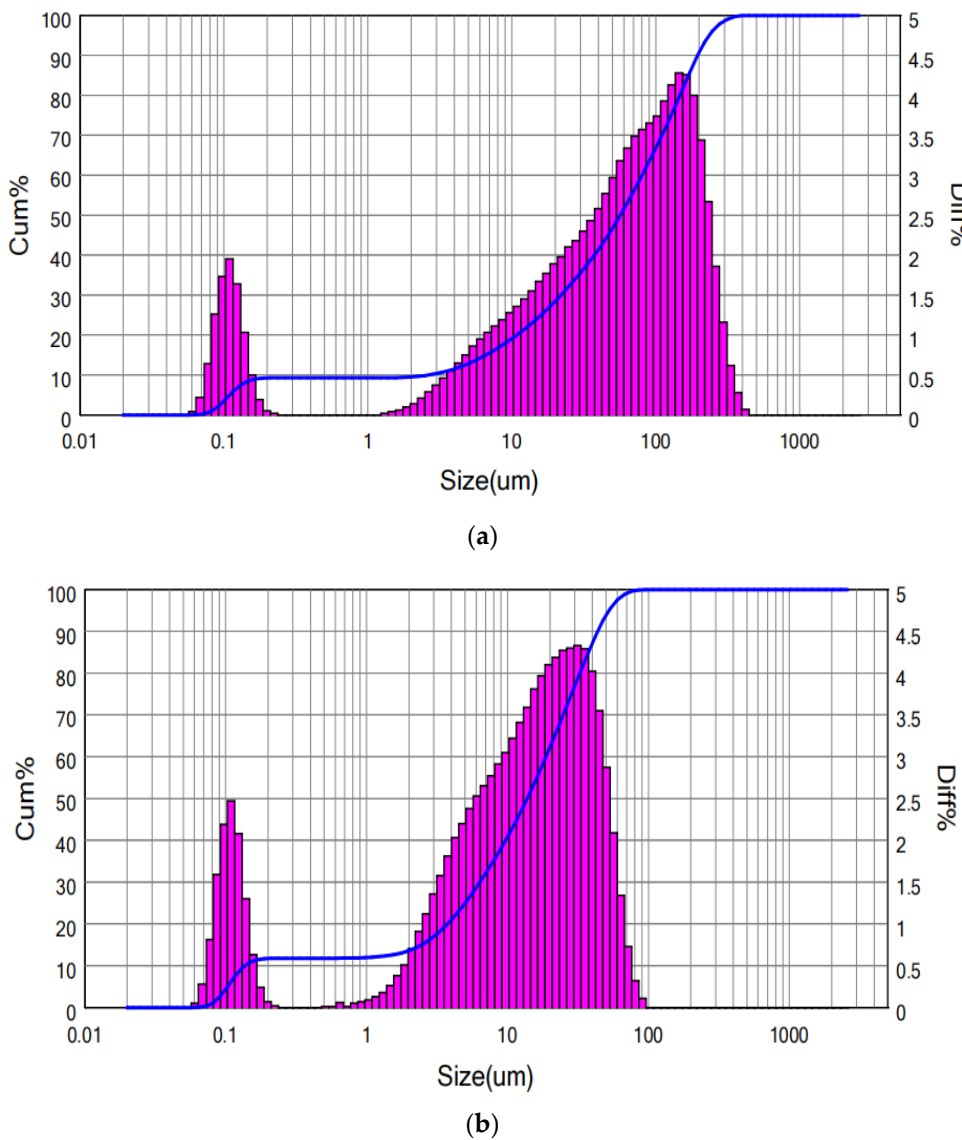

**Figure 1.** Particle size distribution graphs for (**a**) ultrasonically prepared and (**b**) conventionally prepared NiFe$_2$O$_4$ catalysts at 5 wt% Ni loading.

2.1.2. X-ray Diffraction Analysis

The phase purity and crystal structure of synthesized NiFe$_2$O$_4$ catalysts were established using the XRD diffraction analysis. The obtained XRD plots of conventional and ultrasound-assisted synthesized catalysts are depicted in Figure 2.

The prominent peaks were observed at 2θ = 18.51°, 30.98°, 36.01°, 37.17°, 43.68°, 54.21°, 57.69°, 63.35°, and 75.66° with planes as (111), (220), (311), (222), (400), (422), (511), (440), and (533), signifying the cubic spinel structure of the NiFe$_2$O$_4$ complex [20–22]. XRD spectrums confirmed that the novel ultrasound-assisted synthesis method does not alter the structure of the NiFe$_2$O$_4$ complex since both spectra have the same planes. When compared to the conventional synthesis approach, the peak intensities of the ultrasonically synthesized catalyst complex were found to be greater, indicating the creation of crystalline particles with higher crystallinity. Besides this, the absence of any intermediate additional peak revealed the formation of a highly pure catalyst complex. In general, it was observed that ultrasound facilitated the formation of an effective catalyst, especially with respect to crystalline content and lower particle size, which can offer better results in the catalytic reaction. With an objective to confirm this hypothesis, actual degradation studies for

Chrysoidine R dye were also performed, providing a detailed understanding into the effect of operating conditions.

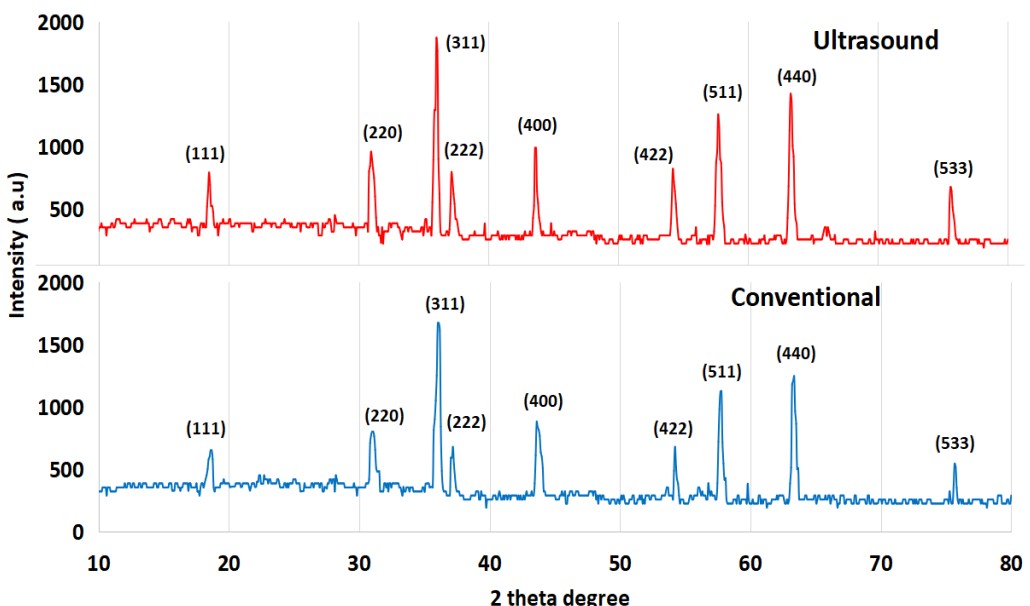

**Figure 2.** XRD spectrum of ultrasonically prepared and conventionally prepared $NiFe_2O_4$ catalysts at 5 wt% Ni loading.

### 2.2. Ultrasound-Assisted Chrysoidine R Dye Degradation Studies

2.2.1. Effect of pH on the Degradation of Chrysoidine R Dye

　　pH is a crucial factor to be considered while studying dye degradation as it is widely reported to influence the degradation rates. In the present study, the pH of the 20-ppm dye solution was varied between 3 to 9. The degradation studies were performed using 0.4 g/L catalyst loading (ultrasonically synthesized) under conditions of 40 °C, time duration of 180 min, ultrasound power of 100 W, and duty cycle of 70%. The obtained results are plotted in Figure 3a in terms of the extent of degradation and 3b in terms of kinetic studies.

　　It was observed that degradation increased when the solution was made acidic. At 9 pH, the maximum Chrysoidine R dye degradation observed was only 5.08%, which increased to 15.06% and 34.03% when the acidity of the dye solution increased to pH of 5 and 4, respectively. Furthermore, the degradation showed an increasing trend as the acidity of the dye solution further increased from pH of 4 to 3. At 3 pH, maximum dye degradation of 39.49% was noted, and pH 3 was chosen as the best operating condition for further experimental studies, keeping in mind the corrosion and life of the ultrasonic horn (still lower pH were not investigated in the work). The observed trend of higher degradation under acidic conditions can be attributed to the presence of a pollutant in the molecular state instead of the ionized state. In the molecular state, hydrophobicity shifts the molecules towards the liquid–gas interface where the concentration of •OH radicals is much higher to facilitate direct attack, resulting in higher degradation. In bulk liquid, the •OH radical concentration is less, and hence, lower extent of degradation is seen. In addition, the oxidation capacity of the hydroxyl radicals that are mainly involved in the degradation process is also higher under acidic conditions. It was also seen that Chrysoidine R dye degradation followed second-order reaction kinetics. The obtained kinetic data are tabulated in Table 1.

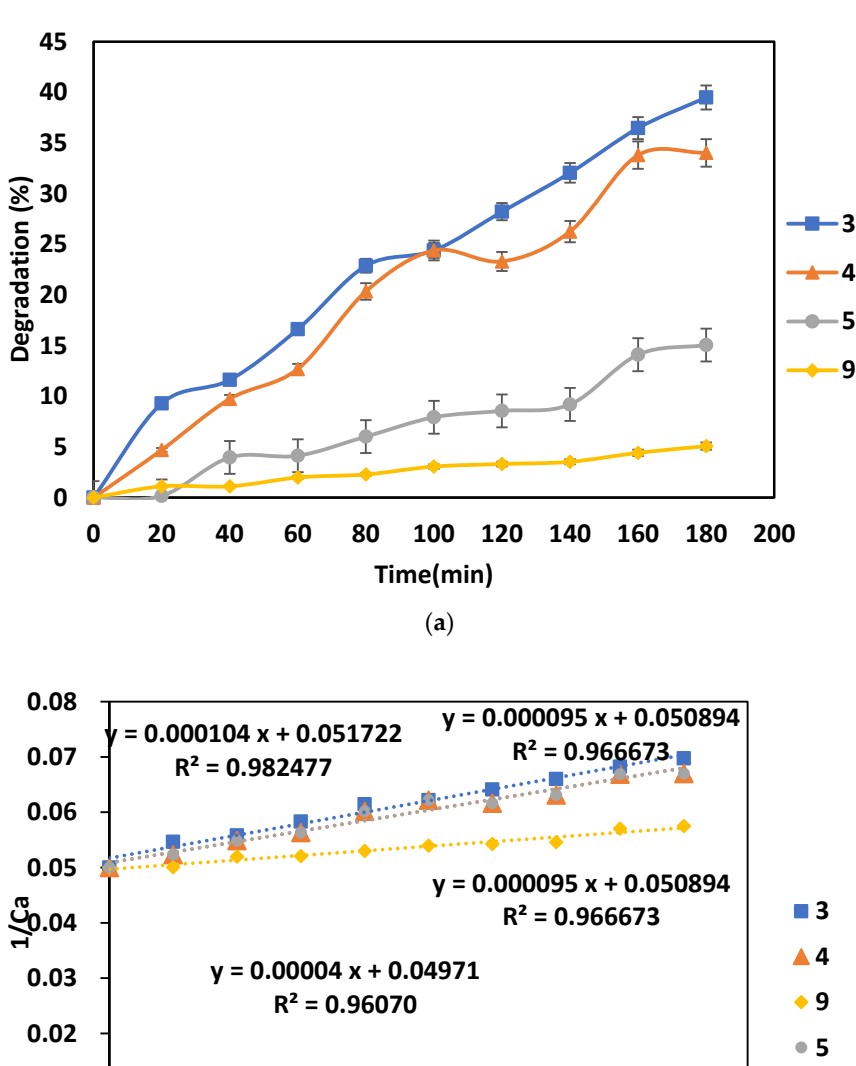

**Figure 3.** Effect of pH on the degradation of Chrysoidine R dye: (**a**) extent of degradation and (**b**) kinetic fitting.

**Table 1.** Extent of degradation and kinetic rate constant data at various pH.

| pH | Degradation (%) | Second-Order Rate Constant $(K \times 10^{-5} \text{ mL}^{-1}\text{min}^{-1})$ | $R^2$ |
|---|---|---|---|
| 3 | 39.49 | 10.4 | 0.9824 |
| 4 | 34.03 | 9.5 | 0.9667 |
| 5 | 15.06 | 9.5 | 0.9667 |
| 9 | 5.08 | 4.2 | 0.9606 |

It was again elucidated from Table 1 that the kinetic rate constant also showed similar trends to that of the extent of degradation variation with pH. Liu et al. [23] studied the rhodamine B dye degradation over the pH range of 2–4.5 and found at pH 3, the degradation

was maximum with a value of 90%. Shiljashree et al. [24] studied Irgalite violet dye degradation using $NiFe_2O_4$ nanoparticles over the range of pH 2.0 to 4.0 and reported maximum degradation of 99.9% was seen at pH 3.0 with lower degradation as 78% for pH of 4. The effect of pH on Remazol Red RB and Direct Green B degradation under the condition of 5 mg/L as nickel ferrite nanoparticle loading, ozone concentration of 55 g/m³, 1 L of dye solution with a concentration of dye as 150 mg/L, and temperature as 25 °C was analyzed by Mahmoodi [25] at pH values of 3, 5, 7, and 9. It was reported that a pH of 3 results in maximum degradation of 86% for RRRB and 90% for DGB. It was clearly elucidated that acidic conditions (best pH of 3) favored Chrysoidine R dye degradation more efficiently and hence was selected for further studies.

### 2.2.2. Ultrasonic Duty Cycle Effect on Degradation of Chrysoidine R Dye

The ultrasound duty cycle represents the on and off time of ultrasonic irradiation. Duty cycle effect on chrysoidine R dye degradation was investigated by varying the duty cycle over the range of 50–80%. The experiments were performed at a constant power output of 100 W, 0.4 g/L as the catalyst loading, 40 °C temperature, 180 min as treatment time, and pH of 3. Figure 4 represents the graphical plot of the obtained results in terms of extent of degradation and kinetic rate constants.

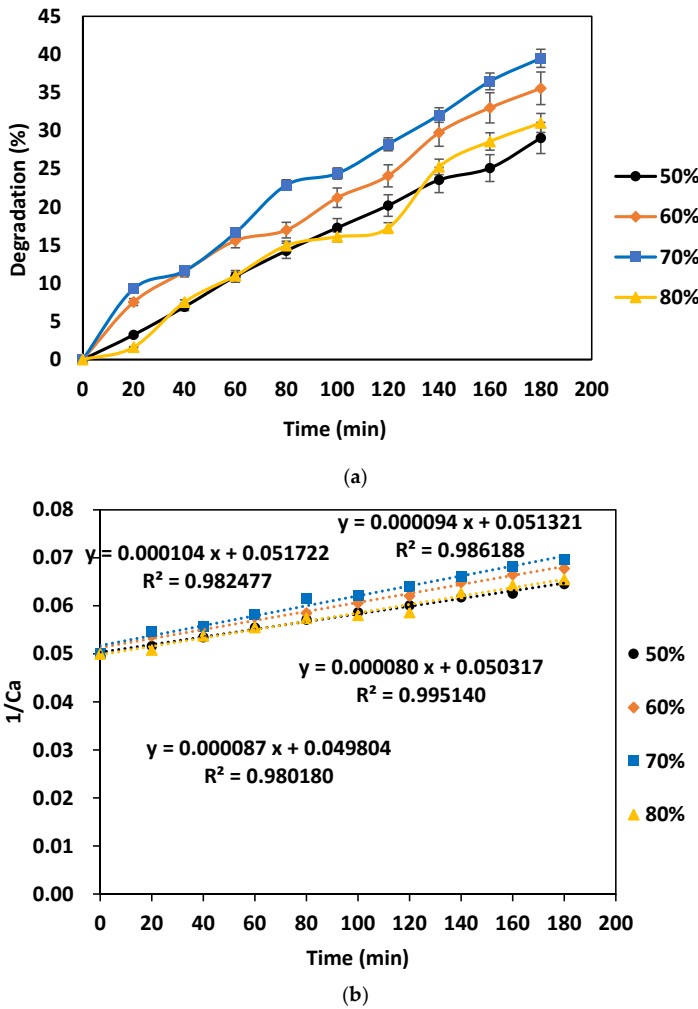

**Figure 4.** Effect of ultrasound duty cycle on the degradation of Chrysoidine R dye: (**a**) extent of degradation and (**b**) kinetic fitting.

It was seen that the degradation almost doubled from 29.05% to 39.48% when the duty cycle increased from 50% to 70%, indicating that the elevation of on time of ultrasonic

irradiation can significantly improve the rate of degradation. This enhancement can be ascribed to the enhanced activity of cavitation resulting in an increased number of active hydroxyl radical generation. It was also seen that a subsequent increase of the duty cycle to 80% gave lower degradation of 35.56%, which can be attributed to the fact that beyond the optimum duty cycle, the number of cavity formations increases too much, leading to a possible coalescence of the cavities, yielding cushioned collapse of much lower intensity giving the lower extent of degradation [26]. The degradation of Chrysoidine R dye was also found to follow second-order kinetics, and the obtained data is presented in Figure 4b. The trend for the kinetic rate constant was also similar to the trend in the extent of degradation. Thakare et al. [27] also reported that COD reduction increased continuously until 60% of the duty cycle, giving the actual value as 21.59%, and a marginally lower value as 20.26% was observed at 80% duty cycle, indicating 60% as the optimum.

### 2.2.3. Effect of Catalyst Dosage

Selecting the optimum quantum of the catalyst is another important requirement in the sono-catalytic oxidation, and hence, the effect of dosage of 5 wt% Ni-loaded $NiFe_2O_4$ catalyst (Ni content was kept fixed during the synthesis) was studied using various values as 0.4, 0.5, and 0.6 g/L. The results shown in Figure 5a elucidate that using 0.4 g/L as the catalyst loading led to 39.49% dye degradation, which increased to about 43% when the catalyst loading was increased to 0.5 g/L.

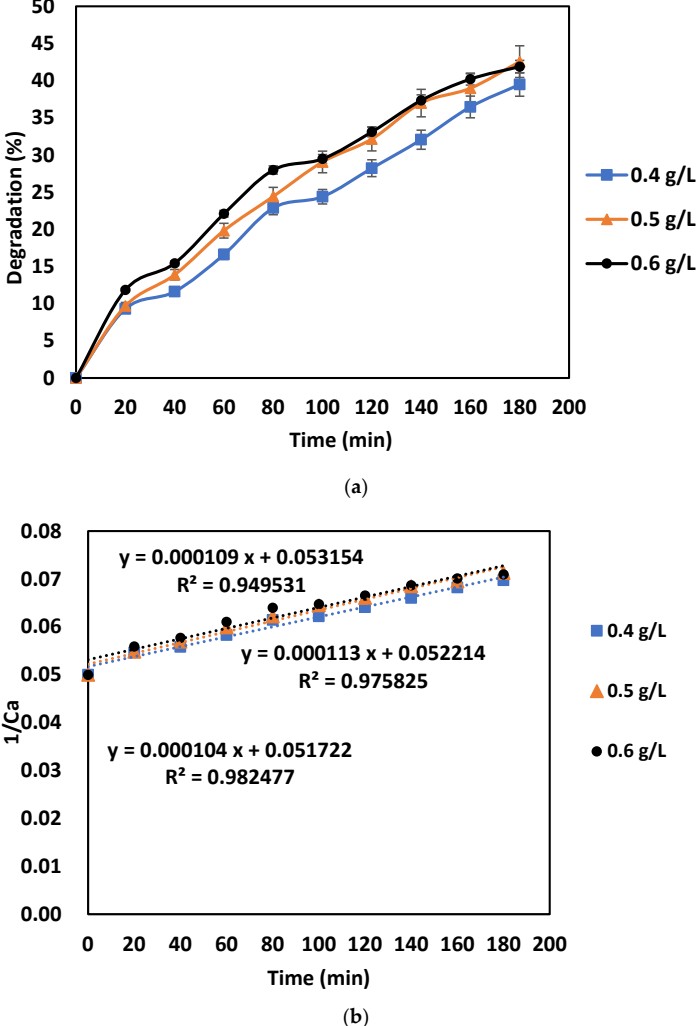

**Figure 5.** Effect of catalyst dosage on the degradation of Chrysoidine R dye: (**a**) extent of degradation and (**b**) kinetic fitting.

The trend was attributed to the fact that the added $NiFe_2O_4$ catalyst till an optimum loading provided additional heterogeneous nuclei for the cavitating bubble formation and hot spots, which subsequently generates more active hydroxyl radicals for dye degradation. However, a further increase in the loading to 0.6 g/L resulted in a marginal reduction in the degradation to 41.89%, as per the data shown in Figure 5a. The trend may be associated with the consideration that an excessive presence of catalyst can scatter the ultrasound, altering the efficient transmission, which reduces the cavitational intensity and hence the rate of sonocatalytic degradation [28]. In addition, the possible explanation would be that the excessive catalyst can interfere with the collapsing bubble, leading to a less powerful collapse [29]. The second-order kinetics was found to fit well to the dye degradation, as depicted in Figure 5b. It was observed that the second-order constant increased from $10.9 \times 10^{-5}$ and $11.3 \times 10^{-5}$ $mL^{-1}min^{-1}$ for a change in catalyst dose from 0.4 g/L to 0.5 g/L, but it decreased to $10.4 \times 10^{-5}$ $mL^{-1}min^{-1}$ at a catalyst loading of 0.6 g/L. Bose et al. [30] also elucidated that increasing the $MgFe_2O_4$ catalyst dosage from 1 g/L to 1.25 g/L showed insignificant improvements in brilliant green dye degradation. Sobana et al. [31] reported similar trends for Direct Blue 53 dye degradation in the presence of an Ag-TiO$_2$ catalyst, whereas Abdellah et al. [32] reported a similar trend for the methylene blue degradation using the $TiO_2$ catalyst. The literature analysis revealed the existence of different optimum loading, confirming the importance of the current planned work for Chrysoidine R dye. Based on the results obtained in the present work, 0.5 g/L was considered as optimum dosage.

### 2.2.4. Effect of Power of Sonication

The effect of the power on the degradation of dye was studied over the range of 80 W to 120 W. The obtained Chrysoidine R dye degradation at different power outputs is shown in Figure 6.

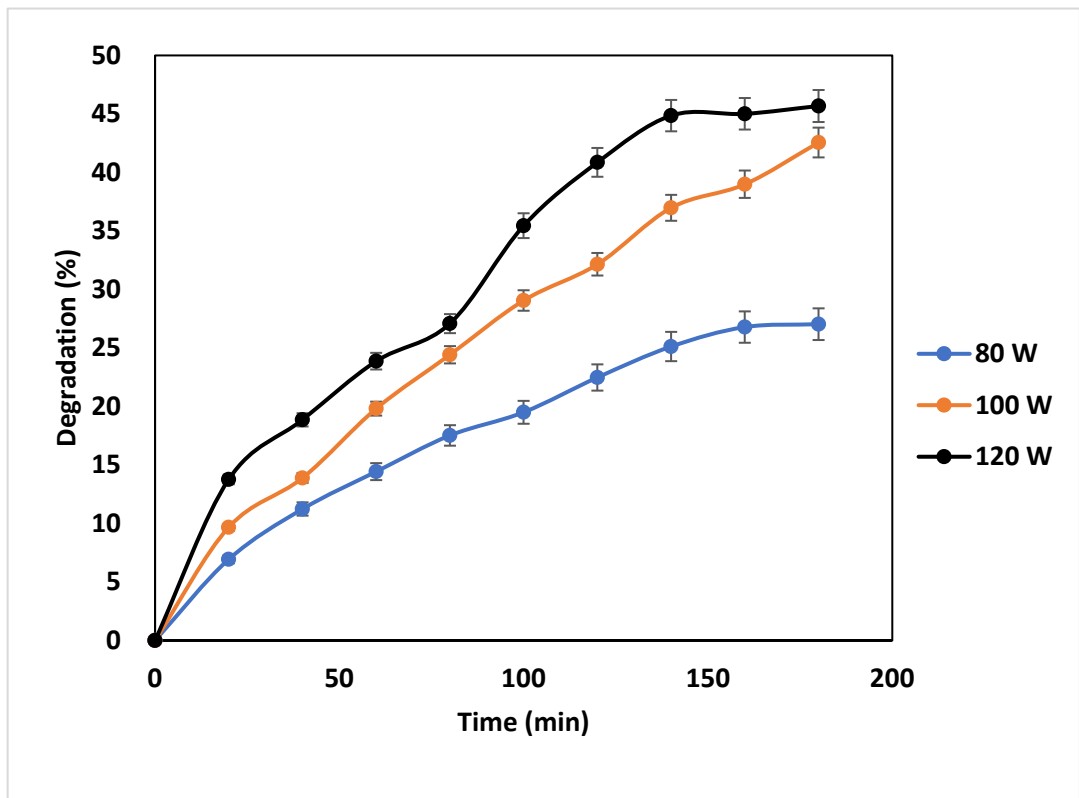

**Figure 6.** Effect of ultrasonic power on the degradation of Chrysoidine R dye.

It was observed that 80 W power resulted in the lowest degradation (27.03%). In comparison to 80 W power, higher degradation of Chrysoidine R dye as 42.56% and 45.68% was observed at 100 W and 120 W, respectively. The increasing trend with increasing power is due to the fact that the use of higher power dissipation results in a higher generation of cavities and active radical formation within the reaction mixture, thereby resulting in higher degradation. A comparable result was reported by Xu et al. [33] where a rise in degradation of rhodamine B from 91.89% to 97.77% was reported with an increase in power from 150 W to 300 W in 40 min of reaction time for 20 ppm rhodamine dye solution. In another study, Kodavatiganti et al. [34] studied the applicability of different power outputs for the degradation of acid violet 7 dye at a loading of 20 ppm. The authors found the optimum power of 100 W resulted in a maximum of 40.1% decolorization of the dye solution. The results obtained in the present work and comparison with the literature showed that the power has a significant effect on the dye degradation efficiency. Considering the operating limit set by the manufacturer and the maximum degradation efficiency obtained, 120 W power output was chosen as optimum for further studies.

### 2.2.5. Comparison of Degradation Using Conventionally and Ultrasonically Prepared NiFe$_2$O$_4$ Catalyst

Experiments were conducted to compare the efficiencies of the NiFe$_2$O$_4$ catalyst prepared using the ultrasound-assisted approach and conventional approach. Both experiments of sono-catalytic degradation were performed under previously optimized conditions of 120 W as the power, 0.5 g/L catalyst loading, 70% duty cycle, and 160 min as the treatment time. The ultrasonically prepared catalyst resulted in higher degradation of 45.68% compared to 29.45% obtained using the conventionally prepared catalyst, as per the data depicted in Figure 7.

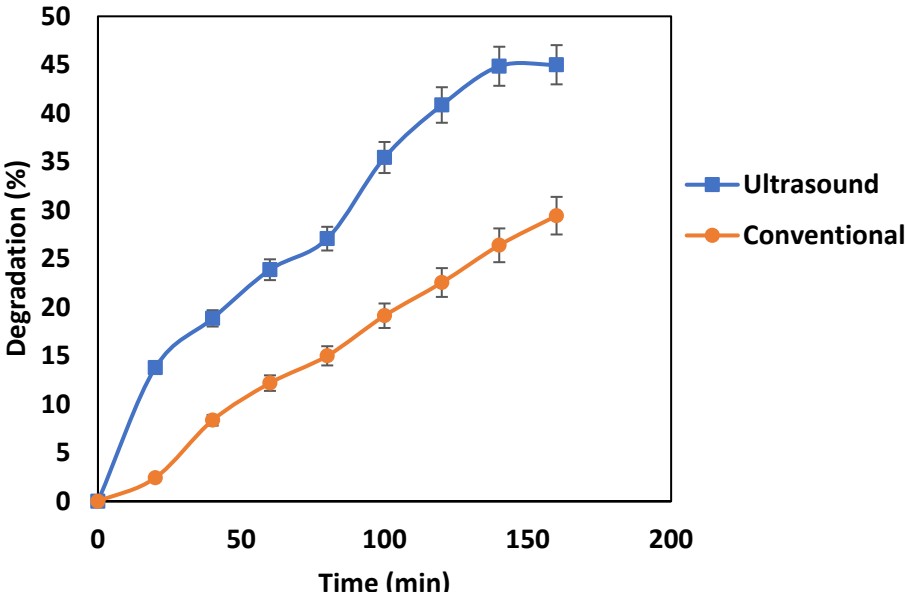

**Figure 7.** Comparison of ultrasonically synthesized and conventionally synthesized catalyst applied for the degradation of Chrysoidine R dye.

The elevated degradation obtained in the case of ultrasonically synthesized catalyst can be accredited to better crystallinity (as per XRD results) and lower particle size of 18.58 μm that results in increased surface area and availability of active sites for sonocatalytic degradation of dye molecules. Similar outcomes have been reported in the literature, highlighting the benefits of sonochemical synthesis. For example, the degradation of CV dye at an initial dye concentration of 50 ppm with 0.1 g rGO-ZnO-TiO2 nano-composite loading was studied using the conventional and sono-chemically synthesized catalyst, and

it was reported that the catalyst obtained sono-chemically resulted in higher degradation as 87.06%, whereas the conventionally prepared catalyst yielded 72.10% degradation [35]. Sancheti et al. [36] reported similar results for brilliant green dye degradation where it was observed that the NiO-CeO$_2$ catalyst synthesized sono-chemically gave 82% of degradation, whereas the same catalyst obtained conventionally gave only 40% degradation. Satdeve et al. [37] evaluated the performance of an Ag-Zno nano-composite synthesized using ultrasonication and a conventional approach for photo-catalytic decolorization of methylene blue dye at 400 mg/L of catalyst loading. It was found that the sono-chemically synthesized catalyst had a higher decolorization efficiency of 96.2% compared to 89.76% for the conventional catalyst. Importantly, the degree of intensification is different in all these literature illustrations, clearly highlighting the importance of the current work for the chrysoidine R dye degradation.

### 2.2.6. Effect of Using H$_2$O$_2$ Combined with the SonoCatalytic Degradation

Hydrogen peroxide is an oxidant that is extensively used in advanced oxidation processes and has been reported to offer excellent oxidation potential when coupled with ultrasound, especially based on the enhanced hydroxyl radical production. To further enhance the degradation of chrysoidine R, hydrogen peroxide was also incorporated as the oxidant in a sonocatalytic study of dye degradation. The influence of different H$_2$O$_2$ loadings, such as 50, 75, and 100 ppm, has been examined at previously optimized conditions, and the obtained results are illustrated in Figure 8.

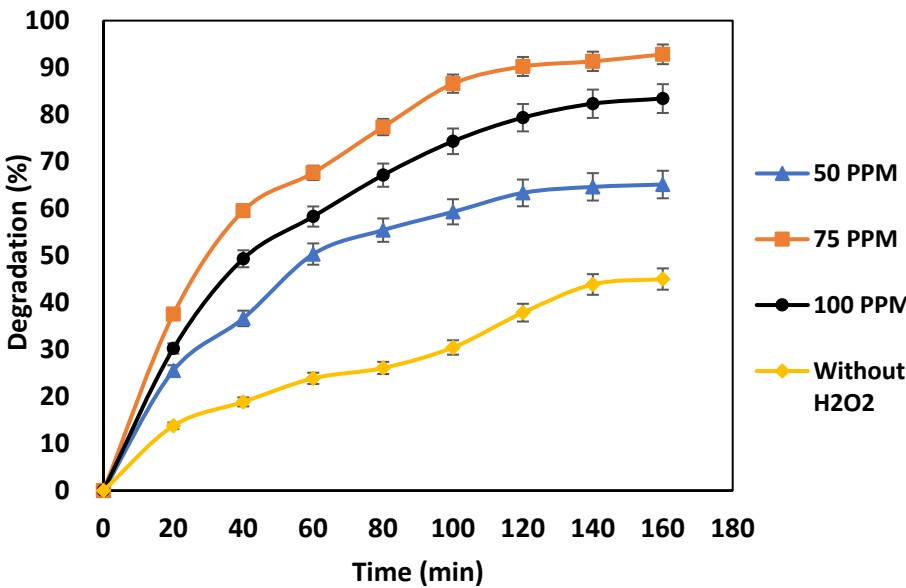

**Figure 8.** Effect of H$_2$O$_2$ addition on the degradation of Chrysoidine R dye.

Without hydrogen peroxide, the degradation observed was around 45%, which significantly rose to about 65.12% with the use of 50 ppm H$_2$O$_2$. A further increase in H$_2$O$_2$ loading to 75 ppm resulted in a further increase in the degradation to 92.83%. At 100 ppm loading, however, there was a reduction in the degradation of dye with an actual value of 83.41%, which is about 11% lower compared to 75 ppm loading. As 75 ppm yielded maximum degradation, it was chosen as the optimum. The declining degradation at the highest oxidant loading can be due to residual hydrogen peroxide, which generally acts as a hydroxyl radical scavenger and favors the generation of per-hydroxyl radical, which is known to possess less oxidation potential, resulting in a decrease in dye degradation [38,39].

$$H_2O_2 + \bullet OH \rightarrow H_2O + \bullet OOH$$

Prakash et al. [40] examined the AB 15 dye removal efficiency at various $H_2O_2$ concentrations ranging from 5 mM to 25 mM and reported that at an optimum of 10 mM loading, 99% dye removal can be achieved, whereas a further increase in $H_2O_2$ concentration to 15 mM, 20 mM, and, 25 mM resulted in lower values of degradation as 98%, 79%, and 75%, respectively. Despite similar trends, the exact optimum concentration value and level of intensification are different, elucidating the importance of the planned work. The presence of the $NiFe_2O_4$ catalyst enhanced the formation of active hydroxyl radials by the following reactions:

$$Ni^{3+} + H_2O_2 \rightarrow Ni^{2+} + H^+ + \bullet OOH$$

$$Fe^{3+} + H_2O_2 \rightarrow Fe^{2+} + H^+ + \bullet OOH$$

$$Ni^{2+} + H_2O_2 \rightarrow Ni^{3+} + \bullet OH + OH^-$$

$$Fe^{2+} + H_2O_2 \rightarrow Fe^{3+} + \bullet OH + OH^-$$

A possible degradation pathway of Chrysoidine R dye in the presence of an active hydroxyl radical has been proposed and illustrated in Figure 9.

**Figure 9.** Proposed reaction mechanism of sono-catalytic degradation of Chrysoidine R dye (* or ● denotes radical).

The breakdown of Chrysoidine R dye begins with the formation of two intermediates: aniline and methylbenzene triamine. Both intermediates follow different paths of degradation. Aniline molecule oxidation results in the formation of benzoquionone, which was broken down into organic acid followed by transformation into carbon dioxide and water. Simultaneously, the methylbenzenetriamine is transformed into methyl benzene triol by liberating water and ammonium cation. Hydroxyl radicals accelerated the conversion of methylbenzene triol to trihydroxybenoic acid, which is further converted to small organic acids, and eventually, the mineralization of small organic acids result in the formation of carbon dioxide and water.

It is important to note that hydroxyl radicals are the main oxidizing species generally observed in the case of ultrasound-induced degradation of contaminants. The presence of hydrogen peroxide at optimum loading further enhances the formation of hydroxyl radicals, which has been confirmed in many of the earlier studies [41,42] based on the use of radical scavengers, such as 2,2,6,6-tetramethylpiperidine-1-oxyl (TEMPO) or hydroquinone (HQ), though not explicitly confirmed in the current work.

## 3. Materials & Methods

### 3.1. Materials

Analytical grade nickel acetate (tetrahydrate), ferric chloride, and sodium hydroxide, which were procured from Molychem Pvt. Ltd., Mumbai, India were used to synthesize the catalyst complex. Chrysoidine R [Basic orange 1 (BO)] dye was purchased from Huntsman Corporation, India. An oxidant, hydrogen peroxide (commercial $H_2O_2$ solution with strength as 30% *w/v*), was procured from Thomas Baker Chemicals Pvt. Ltd. in Mumbai, India. For the preparation of different solutions, distilled water was used, which was freshly prepared using the Milipore distillation unit available in our laboratory.

### 3.2. Experimental Methodology for Catalyst Synthesis

#### 3.2.1. Synthesis of Nickel Ferrite Oxide

The nickel ferrite oxide catalyst complex was synthesized by dissolving the known amounts of nickel acetate tetrahydrate and ferric chloride in 70 mL of water and then mixing the solution with stirring for 30 min. After that, 2 M NaOH was gradually added drop by drop to the homogeneous mixture at a constant stirring rate of 600 rpm until the desired pH of 12 was attained. The resulting pH-balanced solution was then agitated for 2 h at a constant temperature of 40 °C to result in a dark brown color precipitate. The solution was then subjected to sonication (fixed frequency 20 kHz, Dakshin, Mumbai, India) at room temperature. Furthermore, the mixture was washed with distilled water and then filtered with Whatman filter paper. The wet cake was then dried at 80 °C for 6 h and then calcined at 800 °C for 4 h in a muffle furnace. For comparison, the catalyst was also synthesized using the conventional co-precipitation method (without ultrasound) under the same conditions specified for ultrasound-assisted catalyst synthesis, as illustrated in Figure 10. X-ray diffraction was also used as a characterization method to confirm the formation of crystalline $NiFe_2O_4$ catalyst complex as well as the phase purity and crystal structure.

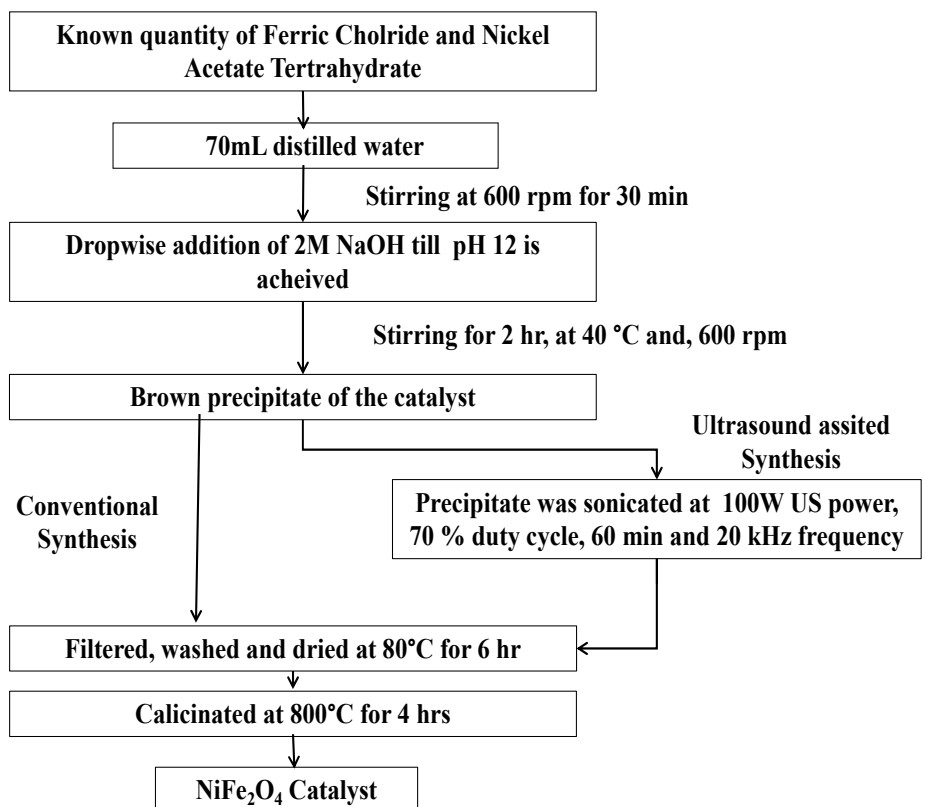

**Figure 10.** Flowchart of conventional and ultrasound-assisted synthesis approaches.

### 3.2.2. Experimental Methodology for Degradation Study

A fixed frequency ultrasonic horn was purchased from M/s Dakshin, Mumbai, India with rated power dissipation as 150 W and a tip diameter of 1 cm. A quartz beaker of 250 mL capacity with a diameter of 68 mm and height of 95 mm was used throughout the experimental study. Figure 11 shows the setup of the sono-catalytic reactor. A well-mixed dye solution (200 mL as fixed quantity in all the experiments) having known concentration kept constant at 20 ppm was used for degradation study. The temperature was maintained at 40 °C using a water bath. The ultrasonic horn tip was dipped at height of 0.5 cm below the liquid level. The process variables investigated were the pH of the solution, duty cycle, catalyst dosage, output power of the ultrasonic horn, irradiation time, and oxidant loading.

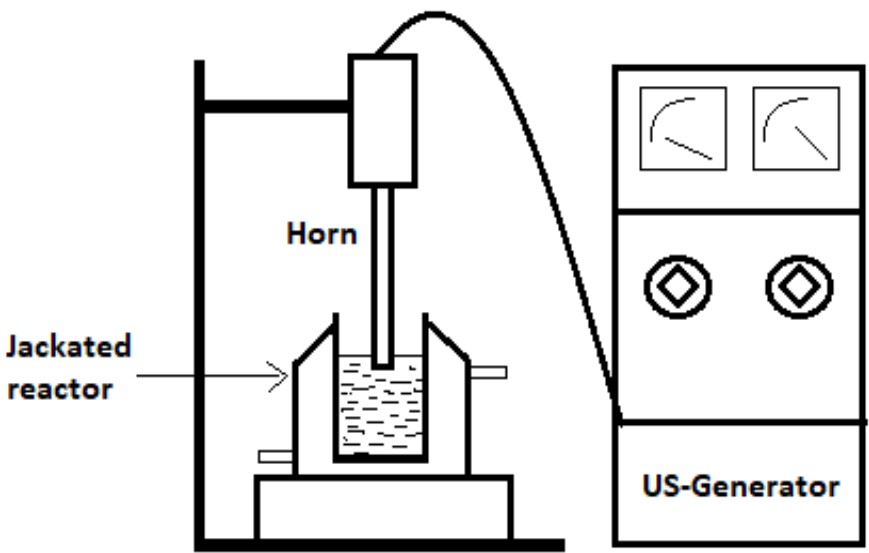

**Figure 11.** Schematic representation of ultrasonic horn used for catalyst synthesis and dye degradation.

### 3.3. UV-Visible Spectroscopy Analysis for Dye Concentration

A double-beam UV spectrophotometer unit (Model UV 1900, Shimadzu, Tokyo, Japan) was used to analyze the samples collected at different time intervals. The wavelength of 452.5 nm was found to have the highest absorbance of Chrysoidine R dye in the visible region. The standard calibration curve was created for dye concentrations ranging from 5 to 25 ppm. The least square method was used for curve fitting, yielding a linear equation of $y = 0.0402x$ with $R^2 = 0.9981$. The percentage degradation was calculated using Equation (1), as shown below:

$$Percentage\ degardation = \left[ \frac{\left( C_i - C_f \right)}{C_i} \right] \times 100 \tag{1}$$

where, $C_i$ and $C_f$ are the initial and final Chrysoidine R dye concentrations, respectively.

### 3.4. Characterization of Nickel Ferrite Oxide Catalyst

#### 3.4.1. Particle Size Analysis

To quantify the influence of the ultrasonic synthesis method as compared to the conventional catalyst synthesis approach, the particle size was analyzed using Bettersizer 2600 E (wet) analyzer.

#### 3.4.2. X-ray Diffraction Analysis

The diffraction patterns of the different catalyst samples were logged using a powder X-ray diffractometer (Philips PW 1800, Phillips, Amsterdam, The Netherlands) equipped with Ni-filtered Cu-Kα radiation (λ = 1.5418 Å) at ambient temperature. The XRD patterns of the sample were recorded at angles between 10° and 80° with a scanning rate of 2°/min.

## 4. Conclusions

The conducted research aimed at the development of an advanced nickel ferrite synthesis method that is more efficient than the current conventional process and at establishing its catalytic performance for the degradation of Chrysoidine R dye. The modification of the conventional method using ultrasound successfully demonstrated that the ultrasound-assisted synthesis method has great potential to improve structural properties and catalytic degradation activity. The study also optimized the ultrasound-assisted catalytic degradation of dye solution using the synthesized catalyst. The results obtained indicated degradation efficiency can be greatly affected by power output and other process conditions. Maximum sono-degradation of 92.83% was obtained under the optimized conditions as 75 ppm $H_2O_2$ loading, pH 3, 70% US duty cycle, 120 W US power output, 160 min of reaction time, and 0.5 g/L ultrasonically synthesized catalyst loading. Thus, it can be concluded that the developed approach of ultrasound-assisted catalytic degradation in the presence of hydrogen peroxide oxidant and sono-chemically synthesized catalyst is an effective sustainable approach for chrysoidine R dye degradation.

**Author Contributions:** Conceptualization, P.R.G.; methodology, Y.M.G. and P.R.G.; investigation, Y.M.G.; data curation, Y.M.G. and P.S.S.; writing—original draft preparation, Y.M.G.; writing—review and editing, P.S.S. and P.R.G.; supervision, P.R.G.; All authors have read and agreed to the published version of the manuscript.

**Funding:** This research received no external funding.

**Data Availability Statement:** Data will be made available on request.

**Conflicts of Interest:** The authors declare no conflict of interest.

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
