# Peer review of "Sonocatalytic Degradation of Chrysoidine R Dye Using Ultrasonically Synthesized NiFe2O4 Catalyst"

_catalysts, doi:10.3390/catal13030597_

Round 1

Reviewer 1 Report

In this paper, the authors report a novel Ni-Fe oxide catalyst that can be employed for dye degradation. The material is novel, while the application is practical. The paper is well organized and well written. Therefore, it can be published after a minor revision:

(1)    References need to be updated. Recent references on iron based catalysts should be cited. E.g. 10.1016/j.cclet.2020.07.031

(2)    The images in the article need to be improved. The authors need to download a published article on materials science and improve their images accordingly.

(3)    In mechanism discussion, The authors supposed a free radical mechanism? This should be supported by experiments, such as the reaction with free radical scavenger (TEMPO or HQ). For examples please see reference: 10.1016/j.cclet.2020.09.012; 10.1016/j.cclet.2022.05.019. Similar reactions can be performed to verify that the free radical mechanism and the related references can be cited to support the hypothesis.

Author Response

Answer: We thank the reviewer for commending on the contents of article for recommending minor revision. We have considered all the comments of the reviewer and suitably modified the manuscript. 

  • References need to be updated. Recent references on iron based catalysts should be cited. E.g. 10.1016/j.cclet.2020.07.031

Answer:- The recent reference has been cited in the manuscript in the introduction section.

  • The images in the article need to be improved. The authors need to download a published article on materials science and improve their images accordingly.

Answer:- The Images all the figures has been updated to high quality in the manuscript.

(3)    In mechanism discussion, The authors supposed a free radical mechanism? This should be supported by experiments, such as the reaction with free radical scavenger (TEMPO or HQ). For examples please see reference: 10.1016/j.cclet.2020.09.012; 10.1016/j.cclet.2022.05.019. Similar reactions can be performed to verify that the free radical mechanism and the related references can be cited to support the hypothesis.

Answer:- The dissociation of hydrogen peroxide in presence of ultrasound and  Fe2+, Fe 3+ into active radicals is widely proposed and reported mechanism. Considering the limited resources we regret not being able to perform the proposed experiments but we have added the related discussion including the suggested references.

Reviewer 2 Report

The authors synthesized the NiFe2O4 catalyst using the ultrasound-assisted co-precipitation method and compared its sono-catalytic efficiency with the conventionally synthetized catalyst, during the degradation of chrysoidine R dye. Various parameters that are relevant for the degradation study were extensively studied. The manuscript is clear and presented in a well-structured manner.

The present study, however, lacks more characterization techniques for the synthesized materials. The two techniques that the authors present (Particle Size Analysis and XRD) do not offer sufficient insight into powder properties. Techniques like SEM, TEM, XPS, FTIR and/or magnetic susceptibility measurement could be added.

1.      The English grammar and style must be improved, many typos and mistakes are found in the manuscript. Please consider rewriting and correcting the following: line 33-35, line 38, line 42 “much high”, line 101-103, line 140-141 “drives/driving”, line 144 “were with”, line 152-155, line 161-162 “in terms”, line 187-188 “drives/driving”, line 355 “that”.

2.      At least two other characterization techniques for the synthesised NiFe2O4 should be added to the study.

3.      More details should be given in the methods section regarding the NiFe2O4 sinthesis?

4.      A higher quality image should be provided for figure 1.

5.      For figures 3-8, different markers should be used for each sample in order to provide clarity.

6.      Error bars should be added to the plots in fig 3-8.

7.     The authors must add the results for reuse cycles (at least 3 cycles) for the most favorable reaction parameters.

8.      Authors should consider studying the formed/ partially oxidized products. If no toxicity study is performed, please include at least some discussion on the topic.

9.      Conclusions should state more clear wich are the optimum parameters for the sono-degradation of chrysoidine R dye (pH value, duty cycle, catalyst loading, etc).

Author Response

 Answer:- The characterization techniques used in the work represents the two important characteristics in terms of the particle size which determines the surface area available for the reaction and also the phase purity and crystal structure using XRD anlysis which is important to decide on the application in wastewater treatment as catalyst. Since the project has ended, we could not implement the other suggested characterization but we will take this as scope for further work.

The English grammar and style must be improved, many typos and mistakes are found in the manuscript.

Please consider rewriting and correcting the following: line 33-35, line 38, line 42 “much high”, line 101-103, line 140-141 “drives/driving”, line 144 “were with”, line 152-155, line 161-162 “in terms”, line 187-188 “drives/driving”, line 355 “that”.

Answer:- The grammar, typos  has been revised in the manuscript. The suggested lines has been carefully rewritten correctly.

  1. At least two other characterization techniques for the synthesised NiFe2O4should be added to the study.

Answer:- Unfortunately as the project has ended and due to availability of limited resources, we are not able to perform additional characterization. However we have now added the particle size data graphs to allow better representation of the obtained particles in addition to only the mean size.

  1. More details should be given in the methods section regarding the NiFe2Osinthesis?

Answer:- To provide more insight  on the catalyst synthesis method a flow chart has been added as Figure 10.

  1. A higher quality image should be provided for figure 1.

Answer:- The Figure 1 image has been changed to higher quality. As per journal formatting changes, it is now represented as Figure 11.

  1. For figures 3-8, different markers should be used for each sample in order to provide clarity.

Answer:- Different markers has been updated in Figure 3 to 8.

  1. Error bars should be added to the plots in fig 3-8.

Answer:- The error bars has been added to the Figure 3 to Figure 8.

  1.    The authors must add the results for reuse cycles (at least 3 cycles) for the most favorable reaction parameters.

Answer:- As the project has ended and due to limited resources, we are unable to perform the recycle study at this point of time. We will definitely consider this as scope for further work.

  1. Authors should consider studying the formed/ partially oxidized products. If no toxicity study is performed, please include at least some discussion on the topic.

Answer:- Since the toxicity study has not been performed, we have added the discussion on the possible oxidized products and decomposition of dye based on our understanding of the reaction mechanism. A scheme has also been proposed as Figure 9.

  1. Conclusions should state more clear which are the optimum parameters for the sono-degradation of chrysoidine R dye (pH value, duty cycle, catalyst loading, etc).

Answer:- The details about optimized conditions has been added in the conclusion section of the manuscript as follows:

Maximum sono-degradation of 92.83% was obtained under the optimized conditions as 75ppm H2O2 loading, pH 3, 70% US duty cycle, 120W US power output, 160 min of reaction time and 0.5 g/L ultrasonically synthesized catalyst loading.

Reviewer 3 Report

Manuscript presented by Yogesh M. Gote et al. shows an experimental study about  the novel ultrasound-assisted co-precipitation method for the synthesis of the NiFe2O4 catalyst. The topic is very important because the present research helps to explain the potential benefits of ultrasound in catalyst synthesis. Thanks to the Authors for conducting this intriguing study.

An already well written and prepared manuscript. Easy to read and follow. Some aspects should be improved. I recommend the article to publish in Catalysts, but first the paper should be corrected. My decision – reconsider after minor revision. Comments to be considered, in order to further improve the manuscript quality:

(1)   Please put citations of Figures, Tabels, ect in main text.

(2)   In section “2.2.1. Synthesis of Nickel Ferrite Oxide” please add information how the structure of synthesised compound has been confirmed.

(3)   Please add information about all differences in synthesis of catalysts: “To quantify the influence of the ultrasonic synthesis method as compared to the conventional catalyst synthesis approach”.

(4)   Please add attachment which confirms the information “3.1.1. Particle Size Analysis”.

(5)   To improve the manuscript, please consider adding a comment about reaction (line 345).

(6)   In the conclusion section please give more details about the research. At the moment, the conclusion is written too much general.

(7)   The Reference style does not match this journal. Correct it.

(8)   Superscripts and subscripts as well as commas and dots should be checked and correct. Avoid extra spaces and enters.

(9)   The English should be improved. Please use British English through manuscript (characterised instead characterized, ect.).

I hope my comments are help for the Authors and the suggestions will contribute to the improvement of presented manuscript.

Author Response

Answer: We thank the reviewer for commending on the contents of article for recommending minor revision. We have considered all the comments of the reviewer and suitably modified the manuscript.

  • Please put citations of Figures, Tabels, ect in main text.

Answer:- The citations of Figures and tables has been carefully included in the main text of the manuscript.

  • In section “2.1. Synthesis of Nickel Ferrite Oxide” please add information how the structure of synthesised compound has been confirmed.

Answer:- XRD analysis was performed in the study to confirm the crystalline nature in terms of the phase purity and crystal structure of the synthesized product as depicted in section 2.2.1 of the manuscript as follows:

X-ray diffraction was also used as a characterisation method to confirm the formation of crystalline NiFe2O4 catalyst complex as well as the phase purity and crystal structure

  • Please add information about all differences in synthesis of catalysts: “To quantify the influence of the ultrasonic synthesis method as compared to the conventional catalyst synthesis approach”.

Answer:- Detailed difference between the method of ultrasonic synthesized catalyst and conventional catalyst has been illustrated in Figure 10 in the section 3.2.1 of the manuscript. The main results have also been highlighted as follows:

Ultrasonically prepared catalyst resulted in higher degradation of 45.68% compared to 29.45% obtained using the conventionally prepared catalyst as per the data depicted in Figure 7. The elevated degradation obtained in the case of ultrasonically synthesized catalyst can be accredited to better crystallinity (as per XRD results) and lower particle size of 18.58μm, that results in increased surface area and availability of active sites for Sono-catalytic degradation of dye molecules.

  • Please add attachment which confirms the information “1.1. Particle Size Analysis”.

Answer:- The particle size graph has been added in section 2.2.1  of the manuscript.

  • To improve the manuscript, please consider adding a comment about reaction (line 345).

Answer:- The proposed reaction mechanism has been added in section 2.2.6 .

  • In the conclusion section please give more details about the research. At the moment, the conclusion is written too much general.

Answer:- The details about optimized conditions has been added in the conclusion section of the manuscript.

Maximum sono-degradation of 92.83% was obtained under the optimized conditions as 75ppm H2O2 loading, pH 3, 70% US duty cycle, 120W US power output, 160 min of reaction time and 0.5 g/L ultrasonically synthesized catalyst loading

  • The Reference style does not match this journal. Correct it.

Answer:- The reference style has been corrected as per the journal requirements.

  • Superscripts and subscripts as well as commas and dots should be checked and correct. Avoid extra spaces and enters.

Answer:- The punctuation and extra spaces are thoroughly examined and corrected in the manuscript.

  • The English should be improved. Please use British English through manuscript (characterisedinstead characterized, ect.).

Answer:- The British English has been updated in the current manuscript as per suggestion.

Round 2

Reviewer 2 Report

The authors responded properly to the previous comments and improved their paper accordingly. 

They offered good motivation for the points that could not be considered.

Hence, I recommend that the paper can be accepted in present form.